# CD4+ Cytotoxic T Cells Involved in the Development of EBV-Associated Diseases

**DOI:** 10.3390/pathogens11080831

**Published:** 2022-07-25

**Authors:** Manuel Ruiz-Pablos

**Affiliations:** Faculty of Medicine, European University of Madrid, 28670 Madrid, Spain; manruipa@gmail.com

**Keywords:** EBV EBNA-1, HLA, cancer, CD4+ CTL, autoimmunity, Gp42, DRB1, DQB1

## Abstract

Activated cytotoxic CD4 T cells (HLA-DR+) play an important role in the control of EBV infection, especially in cells with latency I (EBNA-1). One of the evasion mechanisms of these latency cells is generated by gp42, which, via peripherally binding to the β1 domain of the β chain of MHC class II (HLA-DQ, -DR, and -DP) of the infected B lymphocyte, can block/alter the HLA class II/T-cell receptor (TCR) interaction, and confer an increased level of susceptibility towards the development of EBV-associated autoimmune diseases or cancer in genetically predisposed individuals (HLA-DRB1* and DQB1* alleles). The main developments predisposing the factors of these diseases are: EBV infection; HLA class II risk alleles; sex; and tissue that is infiltrated with EBV-latent cells, forming ectopic lymphoid structures. Therefore, there is a need to identify treatments for eliminating cells with EBV latency, because the current treatments (e.g., antivirals and rituximab) are ineffective.

## 1. Introduction

CD4 T lymphocytes are well-known for their helper roles in pathogen elimination, assisting innate immune responses, B cells, and CD8 T cells. However, they can also perform cytotoxic functions and induce the apoptosis of target cells [1,2]. Cytotoxic CD4 T lymphocytes (CD4 CTLs) show a phenotype (CD4+, CD45RO+, CD28−, CD27−, CCR7−, CD62L−, CCR5+, and CXCR3+) that belongs to the effector memory set (TEM), and with different functional properties to the more classical CD4 T cells that relate them with terminally differentiated CD4 T cells (CD28−) resulting from chronic stimulation (CD27) with antigen experience (memory) [3,4,5]. CD4 TEM cells are multifunctional in terms of cytokine secretion, express high levels of granzyme B and perforin [6], and may be important for protection against certain infections in vivo [7]. The acquisition of cytotoxic activity by CD4 T lymphocytes seems to be regulated by Treg and CD8 T lymphocytes [8], and they generally do not show activation markers (CD38−, HLA-DR−, and CD69−) [5,9], remaining in a stationary state until they are activated (HLA-DR+, CD38+, and CD69+) by their related antigen with strong and repeated activation signals [10], becoming transiently effector T cells with unpredictable phenotypes [11]. In this regard, it has been observed that chronically antigen-stimulated mature CD4 helper T cells can be reprogrammed to become functional CD4 CTLs by inactivating ThPOK expression through a unique mechanism of plasticity at the post-thymic level [8,10,12]. After antigen removal, they “rest” on any of the multiple memory subsets [11]. Thus, the avoidance of activated CD4 CTL (HLA-DR+) is particularly relevant for viruses [8,13], especially for viral infections that infect cells with HLA class II expression, such as the Epstein–Barr virus (EBV) in B cells, or human immunodeficiency virus type 1 (HIV-1) in activated CD4 T cells (HLA-DR+), monocytes, and dendritic cells [9].

### Factors Predisposing towards the Development of EBV-Associated Diseases

The control of EBV-transformed cells is left to adaptive immunity; mainly cell-mediated immunity (CD4 and CD8 T cells) [14]. NK cells are also capable of eliminating cells with type II and III latency [15,16]. During the latency phase, cytotoxic CD8 T lymphocytes (CD8 CTL) that are specific for EBV latency proteins can only eliminate cells with latency II and III by recognizing the latency proteins, LMP1, LMP2A, and LMP2B, that are present on the plasma membrane, and EBNA-3A, EBNA-3B, and EBNA-3C, which are presented on the major histocompatibility complex (MHC) class I of EBV-transformed B cells [15,17]. Hence, MHC class I genes are implicated in an individual’s susceptibility to EBV infection and the development of EBV-associated cancer, as they encode proteins required for the presentation of foreign antigens, such as viral antigens, from the cellular interior to cytotoxic CD8 T lymphocytes [18]. In contrast, CD8 CTLs fail to recognize latency I B cells (only expressing EBNA-1), as EBNA-1 is presented within the MHC class II on these cells [17,19,20,21,22], and, therefore, only CD4 CTLs can recognize this latency antigen [21]. This is where viral glycoproteins (gps) play several important roles in the development of EBV-associated diseases, with entry into target cells being the main role [23]. Five viral gps play an important role during B cell infection: the glycoprotein of the viral envelope, gP350/220, interacts with CD21 or CR2 (the receptor of the C3d component of the complement system) [24,25,26], and gp42 binds to the β1 domain of the β chain of B cells’ MHC class II (HLA-DQ, -DR, and -DP), whereas gB and gH/gL promote membrane fusion [23]. Following EBV fusion with the lipid bilayer of the cell, gp42 becomes peripherally linked to the β1 domain of the β chain of the HLA-DR-DQ or -DP of the infected B lymphocyte, and consequently, alters the HLA class II/T-cell receptor (TCR) interaction [23,27,28], thereby reducing CD4 T-cell activation [29] and EBNA-1 presentation on MHC-II molecules. Thus, the gp42-β1 interaction influences EBV entry into the cell, and it also acts as an immune evasion mechanism by forming a new gp42-MHC-II complex that alters the antigenic presentation to T cells [27,30,31]. This could explain why EBV-associated autoimmune diseases or neoplasms are related to the β DRB1* and DQB1* alleles (Table 1). Since EBV was transferred to a hominid ancestor approximately 12 million years ago, and as the DRB1*04, *03, and *02 lineages are the oldest [32], it may be thought that those individuals with the DR2-DQ6, DR3-DQ2, or DR4-DQ8 haplotypes—against which, the immune evasion mechanisms of this virus have evolved the most—may be less resistant to the infection, and have a greater risk of developing EBV-associated diseases. This would also depend on glutamic acid 46 (E46) and arginine 72 (R72) from HLA class II, which are essential for a stable interaction between Gp2 and MHC-II [33], where E46 is preserved in all HLA-DR,-DP alleles, but only in a small subset of HLA-DQ alleles: β * 02 (β * 0201, β * 0202, and β * 0203) [34]. Therefore, depending on the tissue infiltrated by the B cells with EBV latency in an individual with ancestral MHC-II alleles and an infection of other cell types by the increased expression of MHC-II as a result of increased IFN-γ, one type or another of EBV-associated disease may develop (Figure 1).

Other factors that contribute to the development of autoimmune diseases are sex hormones and sex chromosomes [65,66]. In fact, estrogens decrease the CD4/CD8 ratio, and increase B-cell survival and the release of immunoglobulins [66]. For instance, elevated estrogen levels (estradiol and estriol) during pregnancy have a protective effect by suppressing Th1-responsive autoimmune diseases, but on the other hand, by increasing the survival of autoreactive B cells, they may trigger or enhance Th2-responsive autoimmune diseases [47,65,66]. In autoimmune diseases, there may also be differences in gender and cytokine responses. Whereas men with multiple sclerosis, rheumatoid arthritis, or type 1 diabetes may have an increased Th2 autoimmune response, women may have an increased Th1 autoimmune response [67,68,69,70]. This increase in the survival of self-reactive B cells generated by estrogen in women would raise the chances of the presentation of viral antigens (EBNA-1), thus increasing the risk of presenting viral antigens with cross-reactions to their own antigens, and thus, generating autoimmune responses of a cellular type (Th1).

It is also important to mention that during viral infections, there is a bidirectional cycle between the immune and neuroendocrine systems, where on the one hand, the cytokines released by the innate or adaptive immune response stimulate the release of glucocorticoids, and on the other hand, the glucocorticoids suppress the synthesis and release of these cytokines, protecting the host from damage that is caused by an overactive immune response. In addition, they cause a shift from cellular (Th1/inflammatory) to humoral (Th2/anti-inflammatory) immune responses [71]. Thus, in those individuals that are genetically weak against a virus, where there is poor control of viral infection by cell-mediated immunity, this could lead to a chronic infection, with a continuous innate response. In the case of poor control of EBV, there would be a continuous release of IFN-γ by circulating NK cells and amygdalin CD56bright NK cells to restrict B cell transformation by EBV. However, cell-mediated immunity is needed to eliminate infected cells [15,72]. IFN-γ can be released via adaptive or innate immunity in the antiviral response, and it stimulates the release of glucocorticoids [71]. As such, if there is a continuous production of this cytokine by NK cells in individuals that are genetically weak against EBV, it would then increase the release of cortisol to protect the host from immune damage, and, as a consequence, cause a lower cellular response (Th1) and a shift to the humoral response, further chronicling the infection, and even contributing to the development of autoimmune diseases.

## 2. EBV-Associated Diseases

In this review, we will focus on EBV-associated diseases that develop as a consequence of alterations in CD4 CTL function and activation.

### 2.1. Burkitt’s Disease

Burkitt’s lymphoma (BL) cell lines are latent I (EBNA-1); therefore, they can only be recognized and eliminated by EBNA-1-specific CD4 CTLs [73,74], since EBNA-1 is not present in MHC class I molecules in transformed B cells [73]. In BL, the reduced Th1 (IFN-γ) response of EBNA-1-specific CD4 CTLs may be due to increased levels of IL-10 and Treg cells [75]. Both regulatory T-1 (Tr1) cells [76] and infected B cells [77] may contribute to human IL-10 release by increasing EBNA-1-specific responses in favor of Th2, in BL [75]. These CD4 T lymphocytes with a Th2 cytokine pattern and poor cytolytic activity can sustain primary B-cell infection, and even induce tumor B-cell expansion through IL-4 and IL-13 release, and CD40 contact [8,78,79]. Additionally, BL cells present multiple defects in HLA class II-mediated antigenic presentation, resulting in a decrease in CD4 T cell activation [80], where gp42 could be the cause of the block against HLA-II/TCR interaction by its binding to the β chain of the MHC class II (HLA-DQ, -DR, and -DP) of the infected B cell [23,29]. This might suggest that BL cells with latency I could evade the surveillance of EBNA-1-specific CD4 CTLs in a host with a genetic predisposition (HLA-II alleles) for developing BL, through a decrease in the activation of CD4 T cells by blocking the HLA class II/(TCR) interaction by gp42 and the release of IL-10 by decreasing the Th1 response, increasing the proliferation of Treg cells and the Th2 response. Although the involvement of HLA-II molecules in BL is suspected [80,81], there is currently insufficient evidence to suggest a genetic predisposition for developing the disease based on HLA-II alleles. Therefore, further research is needed to confirm this hypothesis.

Similarly, some children infected by malaria (caused by the Plasmodium falciparum parasite) have demonstrated a decrease in EBV-specific Th1 responses, thus increasing the risk of developing BL [75]. Malaria also promotes the proliferation and increased response of Treg cells, and consequently, an increase in IL-10 and a decrease in the activation of CD4 T cells [75]. This decreases the activation and Th1 response of CD4 CTLs that are specific to other pathogens, such as EBV, increasing the risk of developing BL [75]. In addition, the use of chloroquine (CQ) in regions of Africa against malaria has been associated with an increased lytic replication of EBV, further contributing to the development of endemic BL [82,83,84]. Hydroxychloroquine treatment can increase intracellular pH, and it inhibits lysosomal activity in antigen-presenting cells (APCs), thus preventing processing and antigen presentation in MHC class II molecules to CD4 T cells [85]. Consequently, it reduces the activation and the differentiation of CD4 T cells and cytokines produced by CD4 T cells and B cells (e.g., IL-1, IL-6, and TNF) [85]. Therefore, since EBNA-1 presentation in MHC class II molecules is inhibited by the use of CQ or hydroxychloroquine, B cells with type I latency cannot be recognized by EBNA-1-specific memory CD4 T cells. This, together with a decrease in CD4 T-cell activation caused by CQ, would increase the risk of developing BL.

In early HIV-1 infection, when CD4 T cell levels remain elevated, but CD4 T-cell function begins to be compromised, patients may develop BL [17,86]. This occurs mainly because HIV-1 preferentially infects activated CD4 T cells (HLA-DR and a high expression of CCR5) and resting memory CD4 T cells, causing increased activation (HLA-DR), proliferation, and apoptosis, as the expression of MHC class II molecules in these cells facilitates HIV-1 replication [87,88,89]. Thus, a large percentage of antigen-specific CD4 T cells could die as a result of productive or abortive HIV infection; thus reducing the generation and the maturation of pathogen-specific memory CD4 T cells [90]. It has been suggested that characteristics such as the phenotype, function, and location of CD4 T cells may have a direct influence on their likelihood of being infected and depleted by HIV infection [90]. For example, the susceptibility of Mycobacterium tuberculosis- and cytomegalovirus-specific CD4 T cells to HIV infection has been linked to the specific characteristics of these cells, and different rates of the depletion of these cells could be a determining factor in the timing of reactivation of these pathogens during HIV infection [90]. Thus, although the total number of CD4 T cells is normal during the early HIV-1 infection stage, there is a depletion of pathogen-specific activated memory CD4 CTLs that are needed to control the cells with pathogen latencies (including EBV), which results in an increase of the risk of developing reactivations and malignancies associated with these pathogens [90]. This could be occurring in 55–60% of Burkitt’s lymphomas that are associated with EBV latency I in AIDS patients and in other neoplasms with a higher association [88,91].

### 2.2. X-Linked Lymphoproliferative Disease

The deterioration of CD4 T-cell responses is believed to be responsible for the EBV-induced infectious mononucleosis that is seen in patients with X-linked lymphoproliferative disease, who have a mutation or deletion in signaling lymphocyte activation molecule (SLAM)-associated protein (SAP), an inhibitor of the T cell costimulatory molecule, SLAM or CDw150 [17]. SLAM (or CDw150) is an auto-ligand that is expressed in memory T cells (CD45ROhigh), activated T cells, activated B cells, activated dendritic cells, and activated macrophages [92,93]. It plays an important role in the interaction between T cells and antigen-presenting cells, where it acts as a co-receptor in TCR-dependent responses [93]. Furthermore, it regulates the cytotoxicity of T cells, the induction of interferon-γ in Th1 cells, and the redirection of Th2 clones to a Th1 or Th0 phenotype [94]. In contrast, SAP is expressed in CD4 (Th1 and Th2), CD8, and NK cells, and at very low levels in EBV-transformed lymphoblastoid cell lines (LCL), but not in primary B cells [94]. The SLAM–SAP interaction induces the synthesis of Th2 cytokines, exerts a downward modulation of Th1, and inhibits the activation and expansion of effector T cells [95,96,97]. Consequently, the mutation or deletion of the SAP gene would result in the absence or non-functioning of the SAP protein, and it would lead to abnormal activation of the T cells, and an increased production of Th1 cytokines, as SLAM signaling increases IFN-γ (Th1) secretion and antigen-specific proliferation [92,93,98]. Although patients with XLP present a state of immune hyperactivation, they fail to control EBV infection, resulting in severe and often fatal infectious mononucleosis [99]. In contrast, patients with XLP do not show the same degree of vulnerability to other herpesviruses, such as herpes simplex virus, cytomegalovirus, and varicella zoster, which can cause fatal infections in individuals with other immunodeficiencies [99]. This highlights the unique role of EBV in the pathogenesis of XLP, and the critical role of SAP in anti-EBV immunity [99]. The loss of SAP in T cells leads to altered interactions with B cells, whereas interactions with other APCs remain intact, as SAP potentially regulates T- and B-cell interactions by producing surface proteins or by secreting cytokines necessary for B-cell development [99,100,101]. As EBV predominantly infects B cells, these are the main lytic and latent antigen-presenting cells from EBV to CD8 and CD4 T cells [99]. However, they can be cross-presented by dendritic cells (DCs). Therefore, CD8 and CD4 T cells that are deficient in SAP are not activated when infected B cells specifically present the viral antigen on MHC class I or MHC class II molecules [99]. Although EBV-specific memory CD8 and CD4 CTLs have been formed through cross-presentation by DCs, they would fail to recognize and eliminate EBV-transformed B cells [99]. However, this does not affect the cellular responses of CD8 and CD4 T cells against other infections such as cytomegalovirus or influenza, as these pathogens do not predominantly infect B cells, and their antigens are predominantly presented by other APCs (monocytes and dendritic cells) [99].

Since the elimination of latent I B cells (EBNA-1) requires the activation of EBV-specific memory CD4 CTLs that are capable of recognizing EBNA-1 presented on MHC class II molecules after macroautophagy in EBV-transformed B cells [21], the absence or non-functioning of the SAP protein in these T cells would alter the interaction between the EBV-transformed B cell and the cytotoxic CD4 T cell, preventing the recognition/activation of the CD4 CTL and the elimination of the EBV-transformed B cell. This would increase the risk of developing EBV-associated malignancies in patients with XLP. Therefore, XLP patients who survive primary EBV infection may develop B-cell lymphoma [99]. It has been suggested that in both X-linked lymphoproliferative syndrome and sporadic hemophagocytic lymphohistiocytosis, EBV LMP1 inhibits the expression of the SAP gene, causes the abnormal activation of T lymphocytes, upregulates Th1 cytokines (IFN-γ), and increases macrophage activation, which can lead to the development of hemophagocytosis [98].

### 2.3. Systemic Lupus Erythematosus

Systemic lupus erythematosus (SLE) is a complex autoimmune disease with multi-organ inflammation and varied clinical presentations [102,103]. It is characterized by a loss of self-tolerance with the activation of self-reactive B- and T cells, leading to the production of autoantibodies and tissue lesions [102]. EBV is one of the environmental factors linked to the development of SLE in genetically predisposed patients, since EBV antigens exhibit structural molecular mimicry with common SLE antigens [103]. Patients with SLE have leukopenia (including lymphopenia) [104,105], increased EBV viral load in PBMC compared to healthy controls [103,104], and express higher levels of the four viral mRNAs (BZLF-1, LMP-1, LMP-2, and EBNA-1) in PBMC compared to immunocompetent EBV carriers and patients with infectious mononucleosis [106,107]. This, together with increased levels of IgG antibodies against early diffuse EBV antigen (EA/D) [108] in patients with SLE, indicates the reactivation of the virus. Given the increased expression of the three latent-state mRNAs (LMP-1, LMP-2, and EBNA-1), this also suggests an increase in the cells with EBV latency [106,107]. Thus, they present a defective control of cells with EBV latency due to impaired function (a lower production of IFN-γ) of cytotoxic EBV-specific CD8 T cells [104,109], although Kang et al. observed an increase in EBV-specific CD4+ CD69+ (activated) T cell responses compared to controls [104]. However, Cassaniti et al. observed that EBV-specific CD4 and CD8 T cells had a lower production of IFN-γ in response to viral antigens, compared to controls [110]. These differences in the response of EBV-specific CD4 T cells in patients with SLE may be due to the use of different periods of stimulation (6 h versus 24 h between Kang et al. and Cassaniti et al.) and/or the EBV-specific antigens used [104,110]. Additionally, Draborg et al. reported a significantly reduced amount of activated (CD69+) T cells (CD4 and CD8) and the production of IFN-γ after ex vivo stimulation with EBNA-1 or early diffuse antigen (EBV-EA/D) in SLE patients, indicating that SLE patients have less EBV-specific T cells compared to controls, or that the EBV-specific T cells of SLE patients are not activated after stimulation with EBV antigens [111]. Draborg et al. did not observe that the activation of T cells was the result of a general T-cell defect, since they were activated after stimulation with the SEB superantigen [111]. Contrarily, Cassaniti et al. demonstrated a significantly lower T-cell response to non-specific mitogen (NSM) in patients with SLE compared to healthy subjects [110]. There is also an inverse relationship between EBV EA-specific T cells and the disease activity of SLE patients, and between EBV-specific T cells (EBNA-1 and EA) and the antibodies directed against EBV (EBNA-1 IgG, and IgG EA) [111]; that is, SLE patients have high EBV antibody titers, but very few EBV-specific lymphocytes, whereas controls show high levels of EBV-specific T cells and few EBV-directed antibodies (except for EBNA1 IgG) [111]. Activated EBV-specific CD4 T cells with a Th1 response are essential for the control of EBV infection, and this is also reflected in SLE patients, where higher viral loads are associated with lower numbers of EBV-specific CD69+ CD4+ T cells producing IFN-γ [104]. Thus, there is a poor level of control of latent EBV infection in SLE, with a shift of immune reaction to a humoral response (Th2) in an attempt to control viral reactivation [111]. In general, the overproduction of Th2 cytokines promotes the hyperactivity of B cells and humoral responses, and a decrease in the specific antiviral Th1 response [112]. Patients with SLE have a deficient production of IL-2 by T lymphocytes, and an increased Th2 response, due to increased levels of IL-10 [112,113,114,115,116,117]. However, in SLE, there is also a self-reactive response of CD4 Th1 cells, as occurs in lupus nephritis, where both the humoral and cellular self-reactive responses are involved [118]; that is to say, both of the self-reactive responses are present, but the Th2 humoral response prevails over the Th1. Likewise, patients with SLE show a generally poor immune response with respect to the healthy controls, suggesting that SLE disease exerts an immunosuppressive action [110]. This supports the fact that the reactivations of latent pathogens, such as parvovirus B19, cytomegalovirus, or EBV, are increasing and producing outbreaks of the disease [119]. It even supports an increase in the risk of cancer, e.g., non-Hodgkin’s lymphoma, in patients with SLE [120,121,122]. The immune response of SLE is similar to the Th2 response in lepromatous leprosy, since *Mycobacterium leprae* can not only generate symptoms that mimic lupus outbreaks, including the production of autoantibodies, but they can also act as a trigger for lupus reactivation [119,123]. There are several clinical manifestations of leprosy, including lepromatous leprosy, which is characterized by low antigen-specific cellular immunity, but a high humoral immunity (Th2), and tuberculoid leprosy, which is the result of a high cellular immunity with a Th1 response [124,125]. Lepromatous leprosy exhibits an impaired cellular response to *M. leprae*; a significantly reduced T-cell response to mitogens, such as phytohemagglutinin; and increased antibody levels to *M. leprae* antigens [126,127]. Additionally, as in the SLE, there is an increase in the levels of antibodies against EBV that suggests a deteriorated cellular response against EBV as a consequence of the infection by *M. leprae* [126,127].

The autoimmune response in SLE may be due to the EBNA-1 antigen showing molecular mimicry with common lupus antigens (including Ro, Sm B/B′, and Sm D1), because the EBNA-1 antibodies cross-react with dsDNA or with the spliceosomal protein, Sm [103,104,128,129,130]. Therefore, immunization with EBNA-1 leads to the generation of antibodies against dsDNA or Sm in patients with a genetic predisposition to develop SLE. The haplotypes associated with SLE are those containing the allele HLA-DRB1*15:01 (HLA-DRB1*1501–HLA-DQA1*0102–HLA-DQB1*0602) [35,36,37,38] and the DRB1*03:01 allele (HLA-DRB1*0301–HLA-DQA1*0501–HLA-DQB1*0201), with the HLA-DRB1*1501 allele being of the highest risk [38]. The DRB1*1501 allele also carries the highest risk for leprosy [131,132]. As both pathologies are associated with the infection by an intracellular pathogen that is capable of altering the immune system, they present the same degree of deterioration in the specific cellular response to antigens, and an increase in the levels of antibodies against these pathogens. It seems that the DRB1*1501 allele, together with infection by EBV or *M. Leprae*, predisposes towards the development of SLE or lepromatous leprosy, respectively, with an increase in the Th2 response, and, consequently, an increase in the levels of antibodies.

### 2.4. Sjögren’s Syndrome

Sjögren’s syndrome (SS) is a chronic autoimmune disease that affects the exocrine glands, mainly lacrimal and salivary, generating severe dryness of the eyes and mouth, as a result of lymphocytic infiltration [133,134,135]. The diagnosis of SS is based on the presence of antibodies against the SSA (Ro) and/or SSB (La) antigens, or by means of a biopsy with a characteristic SS pattern [134]. EBV infection has been associated with the development of SS, as salivary gland biopsies from SS patients contain increased levels of EBV DNA compared to normal salivary glands, indicating increased viral reactivation and the inability of the immune system to control cells with EBV latency [133,136]. In addition, the presence of latent infection (EBERs) by EBV has been observed in SS salivary glands forming functional structures that are similar to ectopic germinal centers that favor the in situ activation of B cells and the differentiation of plasma cells, leading to viral reactivation (BFRF1) [137]. The ectopic lymphoid structures of the salivary gland mimic many characteristics of B- and T-cell follicles, allowing a process that is similar to the germinal center to develop [137,138]. This suggests that the ectopic lymphoid structures serve as a reservoir for EBV latency, facilitating the growth and the persistence of EBV through the generation of plasma cells [137,138]. They may even participate in the development of autoimmunity, due to the close association between the production of human anti-EBV IgG antibodies (anti-EBNA-1 and anti-VCA) with the local production of anti-Ro/La, which has been observed in an SS/SCID mouse chimera model [137]. This could indicate an antiviral humoral response that is closely related to autoimmunity, driven by local antigenic stimuli with molecular mimicry to EBV proteins, is present in the ectopic lymphoid structures of SS patients [130,137,139]. Therefore, it could explain the elevated levels of anti-EBV antibodies in the serum of SS patients [140,141]. Interestingly, it has been observed that anti-EBV antibodies (anti-EBNA-1 and anti-VCA) cross-react with SS autoantigens [130,137,139], and that anti-Ro/La autoantibodies precipitate proteins that form complexes with EBER [142]. In addition, EBNA-1 mimics Ro 52 [130], and EBV early antigen D proteins exhibit cross-reactivity with α-fodrin and lipocardin [143].

As in SLE, patients with SS also present a decreased T-lymphocyte response against EBV, along with elevated antibody titers (EBNA, VCA, and EA) against EBV [136,137,140,141]. Although EBV is present in the salivary gland epithelial cells among normal individuals, the salivary gland epithelial cells of SS patients express elevated levels of HLA-DR antigens, which allows them to present EBV antigens to T cells [133]. This may be due to the ineffective control of EBV latency by the T cells of genetically predisposed patients (DR3 and/or DR2) [39,133], which leads to an increase in infected epithelial cells with increased HLA-DR expression to present the viral antigens. This increases the risk of presenting viral antigens with cross-reactivity to self-antigens.

An analysis of the cytotoxic immune response in ectopic lymphoid structures with EBV persistence in SS salivary glands showed an increase in cytotoxic CD4/B-granzyme B-positive T cells that appeared to substitute for cytotoxic CD8 T cells [137,144]. Furthermore, increased levels of cytotoxic CD4 T cells in the peripheral blood have been observed to correlate with an increased infiltration of these cells into the salivary glands, and increased disease activity and severity in SS patients [144]. This could indicate that these cells are behind the cellular autoimmune response against the tissue. It could even be the case that their formation and proliferation take place in the ectopic lymphoid structures of the salivary glands as a consequence of the presentation of autoantigens from the tissue.

Therefore, both the EBV-latent B cells of ectopic lymphoid structures and EBV-infected epithelial cells participate in the development of SS in genetically predisposed individuals by presenting viral antigens with a cross-reaction to self-antigens [137]. This alteration in antigen presentation could be due to the interaction of gp42 with the MHC class II alleles of SS risk. This shows how EBV drives specific deregulation, promotes the survival of self-reactive B-cell clones, and alters tolerance to self. Moreover, this immunological alteration that is caused by EBV avoidance mechanisms in SS patients could explain why they are at greater risk of developing cancer, especially lymphomas [133,136,145].

SS has been associated with the same risk haplotypes as SLE DR2-DQ6 (DRB1*1501, DQA1*0102, and DQB1*0602) and DR3-DQ2 (DRB1*0301, DQA1*0501, and DQB1*0201) [39]. Even the heterozygous genotype, HLA-DRB1*1501–*0301, further increases susceptibility to SS [39,40]. Hence, many SS patients may also develop SLE [146] and autoimmune thyroiditis [147,148,149].

### 2.5. Multiple Sclerosis

Recently, in a longitudinal study, it has been observed that the risk of developing multiple sclerosis (MS) increased 32-fold after infection with EBV, and not with other viruses, thus demonstrating that it is the main risk factor for its development [150]. Furthermore, there is a significantly higher incidence of EBV-induced B-cell transformation in patients with MS compared to healthy subjects, which supports the presence of a higher number of circulating B cells with EBV latency in MS [151,152]. This is also reflected in elevated IgG antibody levels against the EBV-latent protein, EBNA-1, but not against EBV lytic proteins, such as early diffuse antigen (EA-D) [153]. However, other researchers did observe increases in IgG-EA [154,155,156,157] and IgA-EA [158] in patients with MS, suggesting that high levels of IgG and IgA-EA might correlate with disease reactivation and activity [156,158]. There is also an increase in cells with EBV latency that contributes to the inflammatory response in active MS lesions [159]. Therefore, it seems that in MS, there is a poor level of control over the EBV-latent cells, which leads to a rise in the levels of EBV-latent cells and reactivations. Despite this, there is an increased response of CD4 and CD8 T cells to EBV, particularly to EBNA-1, in patients with MS compared to controls [160,161,162,163]. However, it seems that they do not manage to effectively control latent infection, and/or that their increase is the result of the cellular autoimmune (anti-myelin) response. Patients with MS show increased levels and responses of activated CD4 (central memory and effector memory) T cells (HLA-DR) that are specific to EBNA-1 with a Th1 phenotype, which partially cross-react with myelin antigens, compared with healthy carriers of the virus [41,164]. EBNA-1-specific CD4 T cells with myelin cross-reaction produce IFN-γ, but they differ from EBNA-1 monospecific CD4 T cells in their ability to produce interleukin-2, indicative of a polyfunctional phenotype, as found in chronic HIV-1 or controlled EBV viral infections [41]. There are sex differences in the responses of lymphocytes to myelin peptides (myelin basic protein (MBP), myelin oligodendrocyte glycoprotein (MOG), and proteolipid protein (PLP)), where, on the one hand, women with MS show responses that are biased by IFN-γ (Th1), with T cell-mediated demyelination, and on the other hand, men with MS show responses that are biased by IL-5, which may predispose towards more destructive antibody-mediated demyelination [68]. This suggests that men with MS present a Th2 response that probably serves to suppress Th1-dominated responses to IFN-γ [68]. This matches with men with MS having fewer gadolinium contrast-enhanced lesions, but a higher proportion of black holes compared to women, indicating that men with MS are more likely to develop more destructive and less inflammatory lesions than women [165]. Furthermore, unlike the predominant lesion pattern III of MS, with the infiltration of Th1-responding T cells, the pattern II of MS is characterized by antibody/additive-associated demyelination and an accumulation of Th2-responding CD4 T cells that assist B cells in antibody production, along with few Th1-responding T cells, which may also contribute to the humoral response by promoting B cell activation [166,167,168]. Consequently, the Th2-responsive MS pattern II would be more frequent in men with MS. Another study observed that myelin basic protein (MBP)-specific Th2 cells can cause experimental autoimmune encephalomyelitis (EAE), but only in association with T-cell immunodeficiency in RAG-1-deficient mice [168]. This reflects that both self-reactive responses, humoral (Th2) and cellular (Th1), can develop different patterns of MS [166]. It also suggests that the polarization change of a self-reactive response from Th1 to Th2 with treatments would be useless, as it occurred in an experimental model of autoimmune encephalomyelitis induced by MOG, where the immunological deviation of Th1 cells to Th2 after tolerance with soluble MOG led to a higher production of autoantibodies and severe EAE [169].

By pairing patients and controls according to the expression of MS-associated HLA class II alleles, Lunemann et al. suggested that risk alleles, such as HLA-DRB1*1501, predispose towards the selection of cross-reacting EBNA-1 epitopes, and that a higher total number of cross-reacting EBNA-1-specific T cells, generated in a susceptible HLA environment, might contribute to the development of MS [41]. This cross-reaction may be due to the fact that there are two pentapeptic sequence coincidences between EBNA-1 and the myelin basic proteins: QKRPS and PRHRD [170]. In addition, Mescheriakova et al. observed that MS patients had higher levels of EBNA-1 IgG than their healthy siblings and their healthy non-biologically related spouses [171]. Siblings had intermediate levels, and spouses had low levels of EBNA-1 IgG [171]. This suggests a strong genetic contribution to the humoral response to EBNA-1 in MS, associated mainly with the HLA-DRB1*1501 allele [171]. Whether it is typed as Dw2, DR2, DR15, or DRB1*1501, the presence of the 1501 allele of the HLA-DRB1 gene is associated with an increased susceptibility to MS [42]. Therefore, the DR2-DQ6 haplotype containing this allele (DRB1*1501, DQA1*0102, and DQB1*0602) is the highest risk factor, although other haplotypes have been associated with MS, such as DR3-DQ2 (DRB1*0301, DQA1*0501, and DQB1*0201) and DR4-DQ8 (DRB1*04, DQA1*03, and DQB1*0302) [42,54,172,173].

Despite the increased response of EBNA-1-specific CD4 T cells with the Th1 phenotype in most MS patients, Grytten et al. found a 14% increase in cancer risk over the years among MS patients compared to the controls, especially in respiratory organs, urinary organs, and the CNS [174]. Both patients with MS and their siblings had an increased risk of suffering cancer compared to population-based controls. Additionally, the siblings of MS patients demonstrated a higher risk of developing hematological cancers compared to MS patients and controls [174]. Other studies have reported an increased susceptibility for developing Hodgkin’s lymphoma in first-degree relatives of MS patients [175] and among MS patients parents [176]. Several genetic studies have indicated a common mechanism between Hodgkin’s lymphoma and MS, suggesting that genetics and epigenetics are common risk factors for both diseases [174,177]. Therefore, EBV infection in a family environment could be the environmental factor that causes the development of MS or hematological cancer among MS patients that are siblings, since the same epigenetic factors probably regulate both diseases [174,177]. This genetic risk could be the HLA class II alleles (especially HLA-DRB1) of the host, which are associated with EBV infection, predisposing some to develop MS and others to develop hematological cancer in the same family environment. Both MS and Hodgkin’s lymphoma have been associated with the HLA-DRB1*1501, DQA1*01:02, and DQB1*06:02 haplotypes [48,49,50,51,178].

### 2.6. Myasthenia Gravis

Myasthenia gravis (MG) is an autoimmune disease mediated by acetylcholine receptor (AChR) autoantibodies that target the neuromuscular junction, ultimately leading to skeletal muscle weakness and fatigue [179]. The thymus plays an important role in the pathogenesis of MG [180]. Calvalcante et al. showed the persistence and reactivation of EBV in the B cells and hyperplastic thymus plasma cells of patients with MG, suggesting that EBV may contribute to intrathymic B cell dysregulation and altered tolerance in MG patients associated with thymic tumors, probably through the activation and immortalization of self-reactive B cells with EBV latency [181]. Therefore, they associated EBV with B-cell-mediated autoimmunity in MG [181]. Furthermore, the increased expression of CD21 (one of the EBV entry receptors) on the AChR-specific B cells of MG patients supports the contribution of EBV to the activation and expansion of autoreactive B cells in MG [181,182]. An increase in the expression and activation of TLR3 was also observed in patients with MG and thymoma compared to controls, correlating with EBER1 levels [181]. This supports the activation of TLR3 signaling by EBER in MG-associated thymomas. Moreover, TLR3 induces thymic overexpression of the alpha subunit, AChR, and promotes an anti-AChR autoimmune response [180], which supports the hypothesis of an EBV contribution to B-cell-mediated autoimmunity through TLR3 in MG thymomas [181]. Finally, the activation of TLR3 signaling by EBER favors the production of IFN type I and inflammatory responses, which contribute to the recruitment of peripheral B cells [181].

As for the CD4 T cells, most patients with MG have self-reactive CD4 T cells that are specific for AChR, and that probably participate in the synthesis of anti-AChR antibodies through interactions with B cells [183]. These self-reactive CD4 T cells exhibit an inflammatory Th1 response to AChR subunits via the production of the cytokine, IFN-γ [184]. Apart from thymoma, MG has also been associated with an increased risk of developing extrathymal malignancies, such as lymphoid neoplasms [185,186].

Although MS and MG are distinct autoimmune diseases, some studies suggest a comorbidity [187], which could occur due to a common genetic predisposition, provided that both diseases present a higher number of Th1 and Th17 cells, together with their associated cytokines, IL-1, IL-6, IL-17, IFN-γ, and TNF-α [187]. Furthermore, the Treg cells of these patients have numerous dysfunctions [188]. HLA-DRB1*1501 has been identified as the highest-risk allele for late-onset/acquired MG [45,46]. In contrast, the DRB1*0301 allele and the DQB1*0201 allele have been associated with early-onset MG [45,62,63]. Therefore, MS, Hodgkin’s lymphoma, and late-onset MG have a common genetic predisposition: HLA-DRB1*1501, DQA1*01:02, and DQB1*06:02 [45,46,48,49,50,51,178]. However, an environmental factor—in this case, an EBV infection whose gp42 glycoprotein interacts with the domain β1 of the MHC-II chain β—may be necessary for these diseases to develop.

### 2.7. Rheumatoid Arthritis

EBV has been associated with one of the environmental factors contributing to the development of rheumatoid arthritis (RA). As these patients have elevated levels of antibodies against latent and replication proteins (Epstein–Barr viral capsid antigen, early antigen, EBNA-1, and EBNA-2), and are less efficient in neutralizing EBV-infected cells, they are more likely to have significantly higher numbers of circulating EBV-infected B cells, and have an elevated EBV viral DNA load in peripheral blood mononuclear cells (PBMC) compared to controls [23,136,189]. Thus, RA patients are at an increased risk of developing EBV-associated Hodgkin’s and non-Hodgkin’s lymphomas [23,136,190,191]. T-cell-mediated responses to EBV replication cycle proteins and EBV gp110 have been documented in the joint fluid from RA patients [23,189]. Lymphocytes from RA patients also respond poorly to PHA, Candida, and herpes simplex type I (HSV1) antigens [192,193].

EBNA-1 can undergo citrullination, and EBV can generate anti-citrullinated peptide antibodies (ACPA), these being highly specific diagnostic markers for RA [20,189,194,195]. However, citrullination is a generalized post-translational modification of proteins that can occur under physiological conditions and under any inflammatory context, at different anatomical sites [196]. Therefore, citrullination alone cannot explain the onset of citrulline-specific autoimmunity. Some host genetic factors might be involved in the development of RA. It has been shown, both functionally and by peptide-HLA crystal structure determination, that citrullinated peptides are preferentially presented by RA-associated HLA-DRB1 risk alleles [64,197]. These HLA-DRB1 alleles present a five amino acid sequence (R/QK/RRAA) called a shared epitope (SE) [23]. Both HLA-DRB1*0401 and other alleles with the shared epitope, such as *0404 and *1001, can present various citrullinated peptides [64]. Only these MHC class II molecules with the shared epitope have a higher affinity for the citrulline-containing peptide, and as a consequence, this leads to the activation of autoreactive CD4 T cells [197].

In addition, several EBV antigens share similarities with self-antigens; more specifically, the glycine/alanine repeats in EBNA-1 resemble synovial proteins, and the EBV glycoprotein gp110 contains a copy of the shared epitope (SE), leading to an immune response against HLA-DR molecules with the particular shared epitope [189]; that is, antibodies against an EBV-encoded protein (gp110) have sequence homology with QKRAA(SE) of HLA-DR4 [23]. Additionally, it has been proposed that HLAII–gp42 interactions in genetically predisposed individuals with SE-positive DRB1 alleles (DRB1*0401, *0404, *0405, *0408, *0409, *0101, *0102, *1001, and *1402) facilitate EBV entry and infection, which may ultimately result in uncontrolled EBV infection and, consequently, RA onset [23].

The infiltration of memory autoreactive cytotoxic CD4 T cells (CD45RO+) in the synovial joints of patients with RA, together with the genetic association with RA-associated HLA-DRB1 risk alleles, suggests that CD4 T cells are directly involved in the development of RA [64]. These autoreactive CD4 T cells are antigen-experienced (CD45RO+), reactive to citrulline, and they exhibit Th1 response by expressing CXCR3+ [64].

Anti-citrullinated protein antibodies (ACPA) have been described to react to multiple citrullinated peptides from EBNA-1 and EBNA-2 [194,198]. This indicates that EBV proteins may be involved in the generation of the ACPA response [194]. Therefore, EBNA-1, by undergoing citrullination upon its presentation on MHC class II molecules following macroautophagy in EBV-latent B cells [20,189,194,195], is a major candidate in the generation of ACPA- and EBNA-1-specific autoreactive CD4 CTLs in individuals with RA-associated HLA-DRB1 risk alleles. As EBNA-1-specific memory CD4 T cells [198] may cross-react with citrulline in individuals with SE-positive HLA-DRB1 risk alleles, EBV infection coupled with this host’s HLA-DRB1 genetic predisposition may contribute to the pathophysiology of RA by reducing the immune system’s ability to control EBV-transformed cells, causing an increased exposure to EBV antigens, and chronic inflammation [189].

### 2.8. Type 1 Diabetes Mellitus

Type 1 diabetes mellitus (T1D) is caused by insulin deficiency arising from the destruction of pancreatic β cells of the pancreatic islets of Langerhans [47]. Islet antigen-reactive T cells are the main cells that destroy pancreatic β cells, but islet-reactive B cells also play a key role in antigen presentation to T cells, and in the production of cytokines and islet antigen-specific autoantibodies (islet AAb) [199]. Perforin-mediated lysis by cytotoxic T lymphocytes is the main factor necessary for β-cell killing. However, the presence of IFN-γ is also necessary [200], since islet beta cells express HLA class II antigens after their expression is induced by IFN-γ, in combination with tumor necrosis factor (TNF) or lymphotoxin (LT) [201]. The phenotypes of infiltrating CD4 and CD8 T cells are in central memory (T CM, CD45RO+ CD27+) and in effector memory (T EM, CD45RO+ CD27-), with a clear predominance of CD8 versus CD4 T cells [202]. B cells are also found in the pancreatic islets of healthy individuals. Arif et al. described the existence of two different types of insulitis in post-mortem pancreas samples with newly diagnosed T1D: one with a higher number of CD20+ B cells, considered to be a more aggressive form of the autoimmune process, compared to the other type of insulitis with low numbers of B cells [202,203]. Subjects with a higher number of CD20+ B cells have the autoimmune response phenotype (islet AAb++ and IFN-γ >> IL-10), whereas subjects with the other type of insulitis, with low B cell numbers, have the phenotype (islet AAb ± and IFN-γ << IL-10) [203]. IL-10-secreting β-cell-specific CD4 T cells (Th2) have potent regulatory properties, and are present in healthy subjects and relatively enriched in older adults with T1D [203]. Moreover, IL-10-mediated autoreactivity is frequently detected in T1D patients’ siblings who are AAb negative and that have a very low risk of developing diabetes [203,204]; that is, antibody-negative first-degree relatives have a balance of proinflammatory and regulatory T cells, suggesting that even a moderate regulatory response may be sufficient to prevent the development of clinical T1D in genetically predisposed individuals [204]. Indeed, CD4+ CD25+ Tregs that are specific for viral antigens associated with the development of T1D successfully arrest the course of T1D [205]. It has been suggested that β-cell antigens might be similar to these viral antigens, and they could promote the activation of diabetes-preventive CD4+ CD25+ Tregs (specific for these antigens) [205].

However, there is disagreement in the literature as to whether T1D is a Th1- or Th2-mediated autoimmune disease, or both [70]. These discrepancies may depend on factors such as gender, disease duration, and different phases of the autoimmune response [70]. Women with T1D appear to have a higher Th1 response than men [69], and men have higher levels of IL-4 (Th2) than women with T1D [70]. This is similar to what occurs in MS, where men with MS are at higher risk of developing a Th2-responsive MS pattern II compared to women with MS, who show a Th1-biased response [68]. Therefore, there is no protective role of the Th2 response in these autoimmune diseases, and even increased IL-10 in both autoimmune diseases accelerates autoimmune destruction [67,68]. In the case of T1D, IL-10 can promote necrosis through the occlusion of the microvasculature, thereby reducing the viability of larger islets, and it can also promote the Th2 response of CD4 T cells that assist B cells in the production of autoantibodies [67,199]. Lesions in T1D patients with a Th1 response are characterized by an infiltration of T cells (CD8 and CD4), where islet β cells die by apoptosis, sparing the surrounding exocrine tissues [67]. In contrast, the lesions of patients with T1D with a Th2 response are characterized by a paucity of T cells, with an infiltration of eosinophils, macrophages, and fibroblasts, where the islet β cells are killed by necrosis [67]. Thus, both the cellular (Th1) and humoral (Th2) autoreactive responses may be involved in the development of T1D.

The HLA class II haplotypes that confer an increased risk for developing T1D in humans are HLA-DR3-DQ2 or HLA-DR4-DQ8 [55,56]. In contrast, the DRB1*1502-DQB1*0601, and DRB1*1501-DQB1*0602 alleles are negatively associated with type 1A diabetes [47,206]. The association between these HLA class II alleles and the development of T1D indicates that HLA-II-restricted CD4 T cells play an important role in the pathogenesis of the disease [56]. Therefore, with these data, we suggest that in T1D-associated alleles patients, the EBV infection of B cells may result in the presentation of viral determinants that act as peptide mimics to trigger the cross-reactivity of memory CD4 T cells with diabetes-associated antigens [34]. These EBV-infected B cells could influence the activation and proliferation of these autoreactive memory T cells with a Th1 phenotype through chronic exposure to the viral antigen. Both EBV-infected B cells and EBV antigen-specific CD4 CTLs that are cross-reactive with diabetes-associated antigens would infiltrate pancreatic islets. Cross-reactive CD4 CTLs would release IFN-γ, TNF, and LT, inducing the expression of MHC class II molecules on β cells, allowing for the infection of β cells by EBV [34,56]. In HLA-DQ β*02 individuals, it would further increase the infection of cells that express only HLA-DQ, which would otherwise be uninfected in individuals with other HLA-DQ alleles [34]. These cross-reactive HLA-II-restricted CD4 CTLs could directly eliminate islet β cells, promote responses by CD8 CTLs, and could stimulate islet-resident macrophages [56]. Cellular damage and cytokine release as a result of viral infection in a host with T1D-associated HLA-II alleles could lead to an inappropriate immune response, resulting in islet cell destruction and the subsequent development of T1D. The association between EBV with the development of T1D may also explain the widely divergent age of onset among patients with T1D, as this onset may correspond temporally with infection [34]. In addition, there is also an increased risk of developing cancer over the years in patients with T1D [207].

### 2.9. Fulminant Type Diabetes

The DRB1*0405-DQB1*0401 haplotype has been associated with fulminant type 1 diabetes. Both homozygotes and heterozygotes with DRB1*0405-DQB1*0401 show a strong predisposition to fulminant type 1 diabetes [208]. The DRB1*1502-DQB1*0601, but not the DRB1*1501-DQB1*0602 haplotype, has been negatively associated with fulminant type 1 diabetes [208]. Both DRB1*1502-DQB1*0601 and DRB1*1501-DQB1*0602 were negatively associated with type 1A diabetes, but they were not protective against fulminant type 1 diabetes [47,208]. However, Fujiya et al. described a case with fulminant type 1 diabetes mellitus where the disease onset was associated with EBV reactivation that developed in the course of chemotherapy to treat multiple myeloma. This individual had the DRB1*1501-DQB1*0602 haplotype, with no presence of DRB1*0405-DQB1*0401 [47]. They suggested that the EBV evasion mechanisms during the lytic phase in a genetically predisposed host could develop fulminant type 1 diabetes [47]. This is because during the lytic phase, viral interleukin (IL)-10 (vIL-10), which is capable of suppressing the function of Th1 and NK cells, is produced and released, leading to the suppression of interferon-gamma (INF-γ) and IL-2 formation, and the reduced proliferation of CD4 T lymphocytes [47]. This results in an increased differentiation of CD4 T lymphocytes to a Th2 phenotype [47]. The Th2 response is usually predominant during pregnancy, in order to protect the fetus from rejection by the maternal immune system [47]. Many patients have been described to have developed fulminant type 1 diabetes during pregnancy [47]. This indicates that the onset of the disease occurs under the conditions of a predominant Th2 response. An elevated Th2 response is characterized by the absence of the probability of the occurrence of autoimmune diseases with a Th1 response, reduced cell-mediated immunity, and reduced protection against viral infection (reduced virus-specific CTLs) [47]. Therefore, those individuals with HLA class II alleles that are associated with fulminant type 1 diabetes (DRB1*0405-DQB1*0401 or DRB1*1501-DQB1*0602) and an increased Th2 response, as a consequence of EBV evasion mechanisms, may develop the disease upon increased pancreatic β-cell destruction by viral infection [47].

### 2.10. Celiac Disease

Celiac disease (CD) is a chronic autoimmune enteropathy of the small intestine, characterized by an immune system response against deamidated gluten gliadin from the diet in genetically predisposed individuals, generating mucosal surface impairment and, consequently, abnormal nutrient absorption [209,210]. Antigen-presenting cells present deamidated gliadin to CD4 T cells, generating both antigen-specific (Th1) and humoral (Th2) autoimmune responses [209,211]. Th1 cells stimulate CD8 T cells and natural killer (NK) cells, leading to enterocyte apoptosis due to the cytotoxic activity of all three cell types [209,211]. Th2 cells stimulate B cells to differentiate into plasma cells that produce autoantibodies (anti-tTG and antigliadin) [209].

CD occurs in patients with T1D with a prevalence range of 4.4–11.1%, compared to 0.5% for the general population [57]. The mechanism of association of these two diseases involves a shared genetic background: the HLA haplotypes, DR3-DQ2 (DRB1*0301, DQA1*0501, and DQB1*0201) and DR4-DQ8 (DRB1*04, DQA1*03, and DQB1*0302) [55,56,57,58,59]. In addition, it has also been linked to EBV infection [212]. In HLA-DQ β*02 individuals with CD, as in T1D, it would further increase the EBV infection of cells expressing only HLA-DQ that would otherwise be uninfected in individuals with other HLA-DQ alleles [34]. This suggests that EBV could underlie the development of both pathologies in a host with T1D- or CD-associated HLA-II alleles. Consequently, the infiltration of EBV-latent cells, followed by the formation of ectopic lymphoid aggregates in the intestinal mucosa in response to any inflammatory stimulus, leads to an increase in IFNγ [213,214,215]. This cytokine upregulates the MHC class II expression of nearby cells, including enterocytes, allowing for the presentation of self-tissue antigens and deamidated gliadin [213]. Therefore, individuals with CD-associated HLA-II alleles that are susceptible to EBV may present deamidated gliadin as a foreign antigen, which may trigger the development of autoimmune disease. In addition, CD has also been associated with an increased risk of developing cancer, particularly gastrointestinal tract and intestinal lymphoma cancers [216,217].

### 2.11. Autoimmune Thyroiditis

The same could occur in autoimmune thyroiditis (Hashimoto’s thyroiditis (HT) and Graves’ disease (GD)), as they are highly prevalent in patients with T1D [218], and they have been associated with EBV infection [219,220]. A common predisposition has been demonstrated for all autoimmune endocrinopathies, specifically for T1D and thyroid autoimmunity [53]. Badenhoop et al. described that the HLA DQA1*0501 allele was significantly more frequent in T1D (60%), GD (65%), and Addison’s disease (70%) than in controls (43%) [53]. DQA1*0501 is associated with DQB1*02, which is a risk allele in both CD and T1D [221]. In contrast, the DQB1*0602 allele confers protection against T1D and GD [53]. Another study showed that the DRB1*04, DRB1*0301, DRB1*0101, DRB1*0101, DQB1*0201, and DQB1*0302 alleles confer susceptibility to T1D and autoimmune thyroiditis [60]. Particularly, DR3 (DRB1*03-DQB1*02-DQA1*05) has a predisposing effect for GD, and DR7 (DRB1*07-DQB1*02-DQA1*02) has a protective effect for GD [61]. In contrast, the DR4 haplotype is associated with HT [61]. GD, which is characterized by hyperthyroidism caused by thyrotropin receptor (TSHR)-stimulating antibodies, shows an increased Th2 response (IL-4 and IL-10), and thus, increased humoral immunity [222,223]. In contrast, the predominance of cell-mediated immunity and thyroid tissue damage in HT implies a Th1 origin [223]. However, Rapaport et al. suggest that it is incorrect to classify GD and HT as Th1 or Th2 origin diseases, since in both GD and HT, the autoimmune response comprises elements of both responses (Th1 and Th2) [224]. Furthermore, they propose that GD is a stage prior to HT, where thyroid-stimulating antibodies (TSAbs) (Th1-related IgG1) appear in the early humoral autoimmune response, which immediately activate the TSHR and cause hyperthyroidism (GD) [224]. Subsequently, there is a compensatory increase in TSH secretion that maintains the thyroid with a sufficient reserve until it is overwhelmed by massive thyroid follicle destruction and fibrosis, both by cytotoxic T cells (Th1) and by increased levels of antibodies against thyroid peroxidase (TPO) and thyroglobulin (Tg) (Th2-related IgG4), which arise after chronic immune stimulation (over a number of years), resulting in thyroid failure (HT) [224]. Most patients with GD also have autoantibodies against TPO, but they present autoantibodies against Tg less frequently [224]. Additionally, TPO and Tg autoantibodies may comprise IgG4, as well as IgG1 subclasses, implying contributions from both the Th2 and Th1 responses [224].

The haplotype associated with GD is DR3-DQ2, and the haplotypes associated with SLE are DR3-DQ2 and DR15-DQ6. Thus, lupus patients with the DR3-DQ2 haplotype may have an increased risk of developing GD [225,226,227]. However, in SLE, CD4 T lymphocytes present a Th2 response, and Th1 lymphocytes and IFN-γ have been shown to be important for the immune pathogenesis of SLE [225]. Thus, an increase in Th1-responsive lymphocytes in SLE patients is associated with autoimmune thyroiditis [225,226]. In addition, GD can mimic SLE, and both autoimmune diseases can occur in the same patient [228].

## 3. Discussion

There are several hypotheses linking ectopic lymphoid structures with EBV latency to the development of autoimmune diseases and cancer [214,215,229,230,231,232]. For example, these structures have been observed with EBV in the brain of patients with MS, in the synovium of patients with RA, in the salivary glands of patients with SS, in the thyroid of patients with autoimmune thyroiditis, and in the thymus of patients with MG [137,233,234,235,236,237,238,239]. They have also been observed in the kidneys of patients with lupus nephritis, being associated with increased renal function impairment [240]. The same occurs in the pancreas of patients with T1D, where these ectopic lymphoid structures generate autoreactive effector T cells against pancreatic islets, and they may be important for disease progression [241,242]. However, in the latter two cases, the presence of EBV in the lymphoid aggregates was not analyzed.

Ectopic lymphoid structures are aggregates of lymphoid cells that are formed in non-lymphoid tissues due to infectious, autoimmune, or neoplastic processes [214,233,235,243]. Thus, leukocytes circulating in the peripheral blood, including B lymphocytes with EBV latency, are recruited to the affected tissues by inflammatory and antigenic stimuli [214,235,243]. In such tissues, B cells form ectopic lymphoid aggregates that allow for the generation of antigen-specific immune responses [214,243]. This environment is favorable for EBV-latent B cells by using a germinal center-like growth program to transform them into proliferating blasts, and to convert these cells into memory B cells. In addition, antigenic stimuli from this or other microbes, and the help of T cells, allow for the differentiation of plasma cells, leading to viral reactivation [137,214,244,245,246,247,248]. Therefore, this model of ectopic lymphoid structure formation allows EBV to generate viral reservoirs in any tissue with an inflammatory process.

These EBV-latent cells perpetuate the inflammatory state in that tissue by releasing proinflammatory substances, such as EBERs [159,249,250]; by producing abortive reactivations whose proteins can cause inflammation [251,252,253,254,255]; and by releasing new virions that provoke an immune response. Both inflammation caused by the first stimulus (infectious, autoimmune, or neoplastic processes) and inflammation caused by EBV-latent B cells of ectopic lymphoid structures allow for the exposure of foreign antigens [214], leading to the activation of CD4 T lymphocytes and the release of IFN-γ. Likewise, IFN-γ could also be released continuously by NK cells to restrict B-cell transformation by EBV [71]. This increase in IFN-γ levels upregulates MHC-II in adjacent cells, such as epithelial cells, favoring a nonprofessional antigen-presenting cell phenotype [213,256]. Consequently, the newly generated viral particles take advantage of MHC-II expression in order to infect these adjacent cells through the gp42/MHC-II interaction. This is where the different HLA-II alleles could play an important role in the resistance to viral infection and antigenic presentation.

Following EBV fusion with the lipid bilayer, Gp42-MHC-II evasion mechanisms in EBV-latent B cells could give rise to two scenarios leading to the development of EBV-associated diseases: (1) the inhibition of CD4 T-cell activation in response to exogenously-provided, processed antigens and peptide epitopes preventing the recognition and activation of EBV-specific CD4 CTLs, as occurs mostly in EBV-associated malignancies [80]; and/or (2) the induction of activation for viral antigen-specific T lymphocytes that are cross-reactive to self-antigens, which would develop cellular Th1 and/or humoral (Th2) autoimmune diseases [20,41,103,104,128,129,164,170,189,194,195]. Thus, the decreased ability of the immune system to detect and to eliminate EBV latency B cells may be responsible for causing EBV-associated diseases in some patients (Figure 2), such as autoimmunity through the prolonged activation of the immune system [257], and/or cancer through the inhibition of the antiviral-specific Th1 response by an increased Th2 response [75,80]. In EBV-associated autoimmune diseases with a cellular (Th1) and/or humoral (Th2) autoreactive response, there is an increase in EBV latency cells, due to defective control by EBV-specific T lymphocytes [23,111,151,152,189], suggesting that the immune evasion mechanisms of these EBV-latent cells outweigh the surveillance of the T lymphocytes, leading to immunodeficiency. By increasing the number of EBV-latent B cells due to this immunodeficiency by increasing the Th2 response, the presentation of viral antigens would increase, and, consequently, there would be an increased risk of generating viral antigen-specific Th1 cells with cross-reactivity to self-antigens. Hence, in autoimmune diseases, both the Th1 and Th2 responses are present, but one response usually prevails over the other. This could occur in individuals with “ancestral” HLA class II alleles that are associated with the development of these autoimmune diseases (Table 1). EBNA-1 is the main candidate in generating autoimmune responses (Figure 3) because it undergoes citrullination upon presentation on MHC class II molecules after macroautophagy [20,189,194,195], and because it exhibits molecular mimicry with self-antigens [41,128,130,139,170,189], generating IgG against these self-proteins, and/or a specific cellular response. Thus, EBV-associated autoimmune diseases, regardless of the cellular or humoral autoreactive response, could be at an increased risk of developing cancer over the years, as is the case in MS [174] or RA [23,190,191]. Thus, the presence of B cells with EBV latency in ectopic lymphoid structures in the tissue where the autoimmune response develops could lead to the development of B-cell lymphoma, as has been observed in the salivary glands in Sjögren’s syndrome [258,259,260,261], in the thyroid gland in autoimmune thyroiditis [220,262,263,264], and in the central nervous system in MS [265,266]. Even first-degree relatives of patients with MS have an increased susceptibility for developing Hodgkin’s lymphoma, suggesting that the same environmental factor (EBV infection) in a family environment with similar genetic factors (HLA class II) could predispose some to develop an autoimmune disease, and for others, cancer [175,176,177]. This indicates that EBV-associated diseases could develop as a consequence of a previous immunodeficiency caused by this pathogen in genetically predisposed individuals. Both autoimmunity and cancer are not separate entities from immunodeficiency, but they could be interconnected processes [257,267]. This can, for instance, be observed in primary immunodeficiencies, where the development of autoimmune diseases and/or cancer is common [257,267,268]. However, it should be added that the development of EBV-associated neoplasms may not only be associated with the “ancestral” MHC-II alleles, but also with certain MHC class I alleles, since, with the exception of cells with EBV latency I, the rest of the latency types of this virus are controlled by cytotoxic CD8 T cells [15,17,18].

Thus, EBNA-1-specific cytotoxic CD4 T lymphocytes appear to have a protective role in vivo, as deduced from their reduction/absence or altered function in the EBV-associated diseases discussed here, and in other diseases, such as post-transplant lymphoproliferative disease (PTLD) [269] and in EBV-related Hodgkin’s and non-Hodgkin’s lymphomas [15,75,270,271,272,273]. It may be that their “exhaustion” is caused by chronic exposure to viral antigens by participating in the immune response against the virus [214,274,275,276,277], or it may be an effect of the evasion mechanisms of B lymphocytes and macrophages that are infected by EBV [214,249,278,279]. In both cases, an immunodeficiency would develop that cannot control viral latency. However, this would not explain why a virus that is present in 90% of the population [280] does not affect all hosts equally, and is even innocuous in most cases. It is here that the possession of one of the “ancestral” HLA-II alleles, to which EBV has generated resistance, could favor the development of these diseases.

This model supports and completes the hypothesis of autoimmune disease development put forward by Pender [215], in which genetic susceptibility (HLA-II) to the EBV infection of B cells leads to an increased number of latently infected autoreactive memory B cells, which lodge in organs where their target antigen is expressed, and act there as antigen-presenting cells. However, Pender did not discuss the role of gp42 in this susceptibility, nor how the possession of certain ancestral alleles that are related to the B1 domain of HLA-II, where gp42 binds, might favor a greater weakness to infection or altered antigenic presentation. He also failed to mention how Gp42-MHC-II evasion in EBV-latent B cells and IL-10 could inhibit CD4 T-cell activation, with these and other evasion mechanisms being responsible for the immune evasion of EBV latency I cells. Pender posited that the defective control of EBV-latent cells in autoimmune diseases was only the fault of CD8 T cells, this being insufficient to explain the decreased cytotoxicity of T cells, as he also discusses [215]. He further posits that autoreactive B cells are directed to organs containing the target antigen, when these autoreactive B cells could be formed from EBV latency I B cells entering the damaged tissue in response to an inflammatory stimulus, forming ectopic lymphoid structures where antigens from that tissue could be presented as foreign antigens. It could also be due to the increase in non-professional antigen-presenting cells in that tissue via an increase in IFN-γ.

Our hypothesis also supports that of Mangalam et al., where they describe the same ancestral haplotypes raised by us (HLA-DR2DQ6, DR4DQ8, and DR3DQ2), and their role in the development of most autoimmune diseases [281]. However, they did not associate EBV infection as a differential factor between healthy and diseased individuals with the same susceptible haplotypes.

It should be mentioned that this model not only explains the development of the diseases described here, but also others, such as EBV-associated gastric carcinoma, where the formation of ectopic lymphoid structures in the gastric mucosa in response to infection by *Helicobacter pylori* or another inflammatory process causes the infiltration of B cells with EBV latency in the lymphoid aggregates, and the subsequent infection of epithelial cells when expressing MHC-II, due to an increase in IFN-γ [282,283,284,285,286]. Finally, the immune evasion microenvironment generated by these EBV-transformed cells and the clonal growth of EBV-latent epithelial cells would lead to the development of the disease [282,285,286]. This model could even explain the involvement of EBV in the development of chronic fatigue syndrome or myalgic encephalomyelitis [214,278] and long COVID-19 [287]. Both diseases present similar EBV reactivations and chronic symptoms, which could suggest a common EBV immunopathology [278,287,288,289]. In the case of persistent COVID-19, the inflammation caused by the SARS-CoV2 infection of tissues would recruit B cells with EBV latency, where ectopic lymphoid aggregates could form and give rise to viral reactivations. Likewise, it could also help us to understand why the EBV-associated autoimmune diseases, long COVID-19, and chronic fatigue syndrome/myalgic encephalomyelitis are more common in women [290,291,292,293,294,295]. Estrogens, by increasing B-cell survival [66], would allow for a greater permanence of ectopic lymphoid aggregates with EBV latency in the inflamed tissues of patients with “ancestral” HLA-II alleles, and, therefore, a chronification of symptoms. In the case of autoimmunity, this would also favor humoral and cellular autoimmune responses [47,65,66].

Finally, it should be added that EBV-associated autoimmune diseases involve autoreactive cellular (Th1) and/or humoral (Th2) responses, but Th1 to Th2, or Th2 to Th1 polarization would be futile, as it could exacerbate the autoreactive humoral response or the autoreactive cellular response, respectively [169]. In other words, it would fail to cure the autoimmune disease. This suggests that the problem lies with antigen-presenting cells (professional and non-professional) with EBV latency, which are those that promote an autoreactive Th1 and/or humoral Th2 cellular response, presenting viral antigens with molecular mimicry with self-antigens, or presenting antigens from the inflamed tissue itself as foreign. Hence, there is a need to find and to design therapies that can eliminate all types of EBV latency cells, because the current treatments (e.g., antivirals and rituximab) are ineffective (Figure 4C,D) [296,297,298,299].

Therefore, we propose that the use of a DNA demethylating agent (Figure 4A), such as decitabine, in low doses and for a short-course treatment, followed by immunotherapy with EBV-specific CTLs [300] in genetically predisposed individuals (HLA-II) with EBV-associated disease could be beneficial [300]. The prior use of a DNA demethylating agent, such as decitabine, could be useful in transforming EBNA-1-expressing (poorly immunogenic) latency I B cells [21] to latency II and III [300], thus allowing EBV latency B cells to be better recognized and eliminated by self EBV-specific cytotoxic T lymphocytes via immunotherapy. In addition, decitabine would restore HLA-II expression in EBV-transformed LCL [301,302], and would induce lytic infection and apoptosis in EBV-transformed epithelial cells [303,304]. Antivirals (Figure 4A) might also be necessary to prevent virus replication by enhancing lytic infection. Another alternative treatment that could be viable would be the creation of glycoprotein-specific antibodies (gp42 and gp350) that mediate the destruction of EBV-infected cells through antibody-dependent cellular cytotoxicity (ADCC), or the development of an EBV vaccine that enhances the production of these types of antibodies (Figure 4B).

## 4. Conclusions and Future Directions

Future research efforts should focus on better defining those HLA class II alleles with a greater genetic predisposition to EBV infection, and on the development of the diseases discussed here, so that therapies can be sought to completely eliminate EBV in these patients, and, thus, prevent the development of these or other diseases that are associated with this pathogen.

## Figures and Tables

**Figure 1 pathogens-11-00831-f001:**
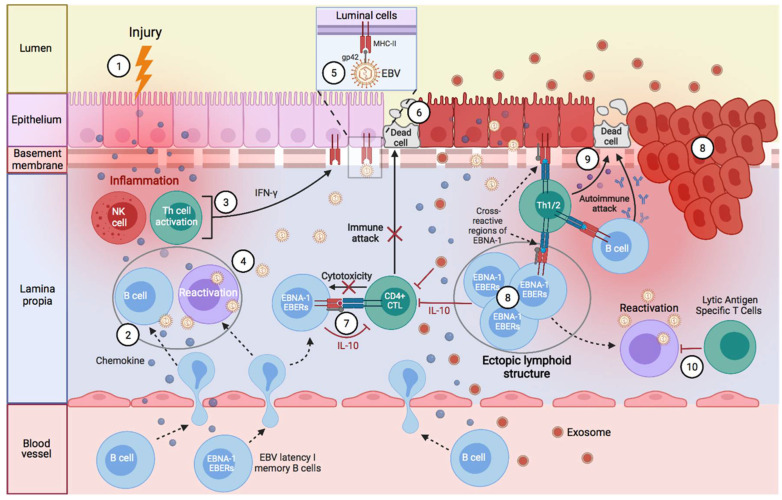
Sequence of EBV infection, chronic inflammation, autoimmunity, and/or cancer in the mucosa of genetically predisposed patients. (1) An infection or any inflammatory stimulus recruit leukocytes in the mucosa, including both latency I B cells (EBNA-1) and healthy B cells. (2) In the mucosa, B cells form ectopic lymphoid aggregates that allow for the generation of antigen-specific immune responses. These ectopic lymphoid structures generate a favorable environment for the transformation of EBV-latent B cells into proliferating blasts, to become memory B cells. (3) In addition, NK cell activation occurs, both in response to the first inflammatory stimulus, and to restrict B cell transformation by EBV. Exposure to foreign antigens from the first stimulus or to viral antigens from EBV leads to activation of CD4 T cells and release of IFN-γ, followed by upregulation of MHC-II on epithelial cells, which favors the acquisition of a nonprofessional antigen-presenting cell phenotype. (4) In addition, the presence of foreign antigens could also lead to terminal differentiation and activation of EBV-latent B lymphocytes, allowing the transition from the latent to the lytic phase of the virus. (5) The newly generated viral particles then infect more epithelial cells through gp42/MHC-II interaction, leading to further inflammation and ultimately to latent EBV infection. Furthermore, this chronic inflammation elicits a cytokine response, leading to increased B-cell recruitment and perpetuation of the viral infection. (6) Latent EBV epithelial cells could enter a lytic phase, releasing new virions, lyse as a consequence of the T-cell response, or undergo neoplastic transformation. (7) The mechanisms of immune evasion of EBV latency (epithelial cells and B cells) involve decreased activation and decreased cytotoxic capacity of EBNA-1-specific CD4 T cells through the release of IL-10 and EBV miRNAs contained in exosomes, which could suppress the expression of target genes in the viral or host genome to maintain latent EBV infection. (8) This altered immunosurveillance leads to increased proliferation of EBV-latent B- and epithelial cells, which increases the risk of neoplastic transformation or autoimmune disease in genetically predisposed patients with EBV-susceptible MHC-II ancestral alleles. (9) Presentation through MHC-II/gp42 of native cellular autoantigens or viral EBNA-1, which can undergo posttranslational modifications, such as citrullination, and form neoantigens, could trigger the activation of autoreactive CD4 T cells and the formation of autoantibodies against tissue cells. (10) Other phases of virus latency or the lytic phase would be controlled by NK cells, and CD4 and CD8 T cells, with specificity for EBV lytic proteins.

**Figure 2 pathogens-11-00831-f002:**
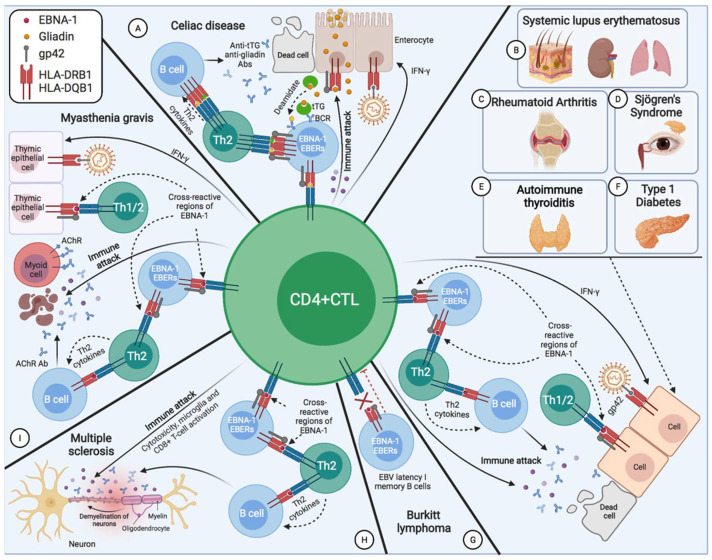
CD4 CTLs involved in the development of EBV-associated diseases. (**A**) In celiac disease, the infiltration of EBV-latent B cells into tissue, and infection of enterocytes via virus transfer from infected B cells increases IFN-γ levels as a response of CD4 CTLs. The increase in IFN-γ increases the expression of MHC-II by enterocytes, further enabling infection of these cells via gp42/MHC-II interaction, fusing the viral lipid bilayer with the cellular lipid bilayer. Gliadin from the diet is internalized into enterocytes by endocytosis, and can cross the epithelium via transcytosis, where they are deaminated by tissue transglutaminase-2 (tTG), thus interacting with antigen-presenting cells. In this case, B cells with EBV latency capture the tTG–gluten complex or deamidated gliadin through the BCR, where they process and present them in MHC-II/gp42 to CD4 T cells, activating them, and generating a cellular autoimmune response against gliadin, and a humoral autoimmune response (Th2) against gliadin and tTG. (**B**) In systemic lupus, erythematosus latency I B cells present EBNA-1 on gp42/MHC-II, activating EBNA-1-specific CD4 T lymphocytes that are cross-reactive to common lupus antigens (Ro, Sm B/B′, and Sm D1) by molecular mimicry, activating Th1 and Th2 cells that generate the autoimmune response. (**C**) In rheumatoid arthritis, as EBNA-1 undergoes citrullination by presentation on MHC class II molecules after macroautophagy in EBV-latent B cells, neoantigens can be formed, activating Th1 and Th2 cells with an autoimmune response against citrulline. (**D**) In Sjögren’s syndrome, EBNA-1 presentation by EBV-latent B cells infiltrating the glands, as well as by infected glandular epithelial cells, activate Th1 and Th2 cells, with an autoimmune response against the glandular epithelial cells, causing cell death and generating severe ocular and oral dryness. (**E**) In Graves’ disease, the infiltration of cells with EBV latency into thyroid tissue increases IFN-γ levels in that tissue as a response of CD4 CTLs, increasing MHC-II expression in thyrocytes, and, thus, allowing infection of these cells by EBV. Both EBNA-1 presentation by EBV-latent B cells infiltrating this tissue, and by infected thyrocytes, activate Th2 cells that are cross-reactive to thyroid antigens. These Th2 cells would help B cells to secrete thyroid-stimulating immunoglobulins against the thyroid-stimulating hormone receptor, resulting in rampant thyroid hormone production and hyperthyroidism. Subsequently, there would be a compensatory increase in TSH secretion that maintains the thyroid with a sufficient reserve until it is overwhelmed by massive thyroid follicle destruction and fibrosis, both by cytotoxic cells (Th1 and CD8) and by the rise of antibodies against thyroid peroxidase and thyroglobulin, which arise after chronic immune stimulation over the years by infiltrating EBV latency B cells, as well as via infected thyrocytes, leading to thyroid insufficiency and, thus, to the development of Hashimoto’s thyroiditis. (**F**) In type 1 diabetes, EBNA-1 presentation by EBV latency B cells infiltrating the pancreas, as by infected pancreatic β cells, activate Th1 and Th2 cells with an autoimmune response against pancreatic β cells, causing cell death and insulin deficiency. (**G**) EBV latency I B cells in Burkitt’s lymphoma are defective in gp42/MHC-II-mediated antigenic presentation, preventing activation and recognition by CD4+ T cells. (**H**) In multiple sclerosis, latency I B cells present EBNA-1 in gp42/MHC-II class II, activating EBNA-1-specific CD4 T lymphocytes that are cross-reactive to myelin, activating Th1 and Th2 cells. (**I**) In myasthenia gravis, EBV latency B cells infiltrating the thymus causes increased levels of IFN-γ released by CD4 CTLs, increasing MHC-II expression in thymic epithelial cells, and, thus, allowing infection of these cells by EBV. Both EBNA-1 presentation by EBV-latent B cells and thymic epithelial cells would activate Th1 and Th2 cells, cross-reacting against the acetylcholine receptor (AChR) by targeting AChR-expressing myoid cells and the neuromuscular junction, ultimately leading to skeletal muscle weakness and fatigue.

**Figure 3 pathogens-11-00831-f003:**
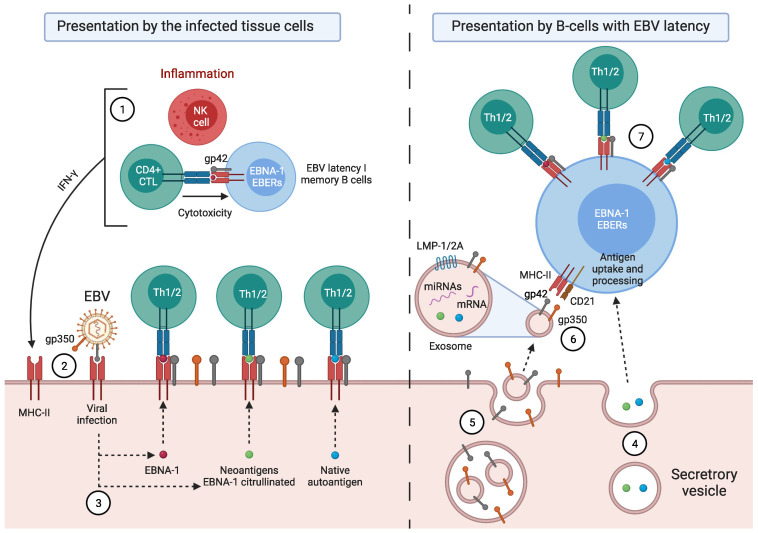
Model of EBV-associated autoimmune disease development. Leukocytes circulating in the peripheral blood, including EBV-latent B cells, are recruited to the affected tissues by inflammatory stimuli (infectious, autoimmune, or neoplastic processes). (1) Both the inflammation caused by the first stimulus and that caused by B cells with EBV latency allow for the activation of CD4 T lymphocytes and NK cells, releasing both IFN-γ. (2) The increase in IFN-γ induces the expression of MHC class II molecules in the cells of that tissue, converting them into nonprofessional antigen-presenting cells and allowing infection through gp42/MHC-II interaction. (3) Through MHC-II, they can present EBNA-1, peptides that undergo post-translational modifications and that can form neoantigens, such as citrullinated EBNA-1, or self-antigens that are native to the cell itself, activating CD4 T lymphocytes and generating Th1 and/or Th2 cells with an autoimmune response against the tissue cells, ultimately causing cell death. (4) Both the neoantigens and native autoantigens can be released by exocytosis, or in exosomes, and be taken up by antigen-presenting cells. (5) Exosomes can have EBV latency proteins, such as LMP-1/2A, and viral glycoproteins, such as gp350 and gp42, in their membrane, since these glycoproteins are present in the cell membrane. Upon internalization of part of the membrane, an endocytic vesicle is formed, which subsequently fuses with the early endosome. After inward budding of the endosome membrane, the intraluminal vesicles will form and give rise to exosomes. They may also contain messenger RNA (mRNA), microRNA (miRNA), and other products of EBV, such as EBV DNA. (6) Exosomes bind to EBV-latent B lymphocytes or other uninfected B lymphocytes through the interaction of gp350/CD21 and/or g42/MHC-II, releasing their contents into the cellular interior. (7) EBV-latent B cells can process and present these antigens in the MHC-II/gp42 complex, activating Th1 and/or Th2 cells with an autoimmune response against the tissue cells.

**Figure 4 pathogens-11-00831-f004:**
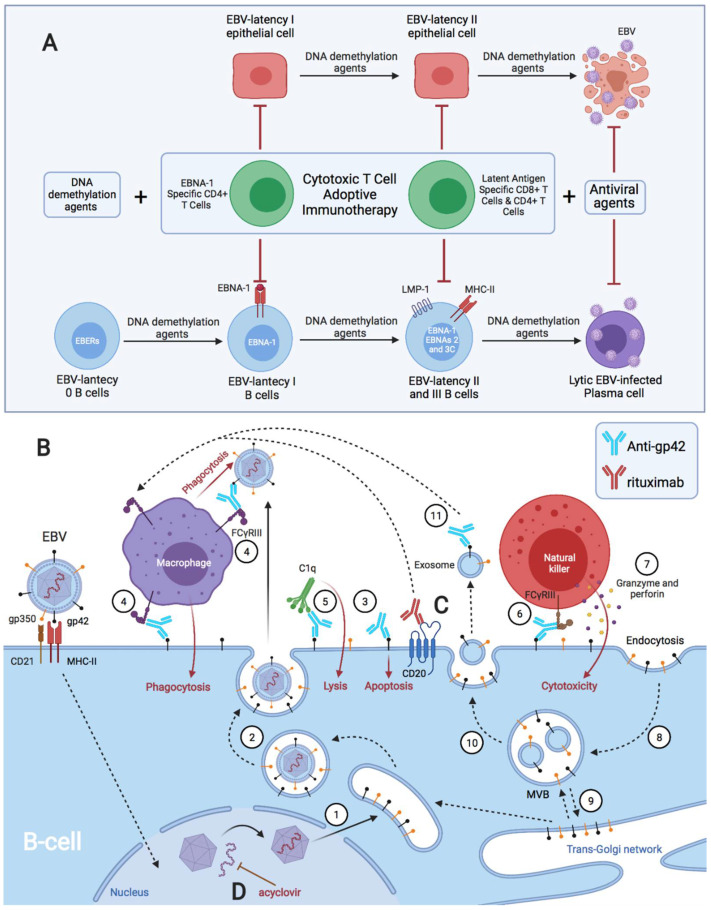
Different treatments against EBV latent cells. (**A**) Schematic model of treatment with DNA demethylation agents followed by adoptive immunotherapy of EBV-specific T cells and antiviral agents. Administration of low-dose DNA demethylation agents restores the expression of MHC class II molecules and induces the expression of LMP1, EBNA-2, EBNA3A, and EBNA-3C, allowing the transformation of EBV latency 0 and I B cells into latency II and III B cells. They also induce the transformation of EBV latency I epithelial cells to latency II. In this way, the recognition of these cells by EBV-specific T cells is improved. EBNA-1-specific CD4 T cells can only recognize latent I cells exhibiting EBNA-1 in MHC class II molecules, since EBNA-1 is poorly immunogenic. DNA demethylation agents induce lytic infection and apoptosis in EBV-transformed B cells and epithelial cells. Antiviral agents prevent viral replication. (**B**) Model of anti-gp42 antibody treatment. (1) Schematic representation of packaging into EBV virions, where the viral nucleocapsid acquires its final lipid envelope by budding in the trans-Golgi network (TGN). (2) It is then transported to the plasma membrane in secretory vesicles and released from the cell. Finally, after this whole process, viral glycoproteins, such as gp42 and gp350, from the secretory vesicles remain in the plasma membrane of the cell. (3) These glycoproteins can be detected by specific antibodies, such as anti-gp42. Following anti-gp42 binding, cells with gp42 on their membrane can undergo: (4) antibody-dependent phagocytosis by activated macrophages, (5) complement-mediated cytotoxicity leading to cell lysis, (6) direct death mediated by natural killer cells, (7) and antibody-dependent cellular cytotoxicity mediated by perforin and granzyme cytokines. (8) Exosome formation begins via endocytosis, where part of the membrane, together with membrane receptors and viral glycoproteins, are internalized, forming an endocytic vesicle, which subsequently fuses with the early endosome. (9) During the maturation process of the early endosome, it communicates with the Golgi apparatus through the exchange of vesicles in a bidirectional manner, forming the late endosome or multivesicular body (MVB). (10) Inward budding of the endosome membrane forms the intraluminal vesicles that will be released into the extracellular space as exosomes. (11) In the case of exosomes with viral glycoproteins present on their membrane, anti-gp42 binding triggers antibody-dependent phagocytosis by activated macrophages. (**C**) Anti-CD20 monoclonal antibodies (rituximab) also act through complement-dependent cytotoxicity, antibody-dependent cellular phagocytosis, antibody-dependent cellular cytotoxicity, and the induction of apoptosis. (**D**) Antivirals prevent viral replication by inhibiting viral DNA synthesis.

**Table 1 pathogens-11-00831-t001:** Main haplotypes related to genetic predisposition to develop diseases associated with EBV.

	EBV-Associated Diseases
DR2-DQ6 (DRB1*1501, DQA1*0102, DQB1*0602)	Positive correlation with: Systemic lupus erythematosus [35,36,37,38] Sjögren’s syndrome [39,40] Multiple sclerosis [41,42] Myalgic encephalomyelitis/chronic fatigue syndrome [43,44] Late-onset/acquired myasthenia gravis [45,46] Fulminant type diabetes [47] Hodgkin’s lymphoma [48,49,50,51] Negative correlation with: Diabetes mellitus type 1 [47,52] Graves’ disease [53]
DR3-DQ2 (DRB1*0301, DQA1*0501, DQB1*0201)	Positive correlation with: Systemic lupus erythematosus [38] Multiple sclerosis [54] Diabetes mellitus type 1 [55,56] Celiac disease [57,58,59] Graves’ disease [60,61] Sjögren’s syndrome [39,40] Early-onset myasthenia gravis [45,62,63]
DR4-DQ8 (DRB1*04, DQA1*03, DQB1*0302)	Positive correlation with: Multiple sclerosis [54] Diabetes mellitus type 1 [55,56] Celiac disease [57,58,59] Rheumatoid arthritis [23,64] Hashimoto’s thyroiditis [60,61]

## Data Availability

Not applicable.

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
