# Peer review of "CD4+ Cytotoxic T Cells Involved in the Development of EBV-Associated Diseases"

_pathogens, 2022, doi:10.3390/pathogens11080831_

Round 1
Reviewer 1 Report
This is a very well written and comprehensive review of EBV associated diseases and the role of CD4+ cytotoxic T cells in modulating responses to EBV. This review also discusses how impaired CD4 cytotoxic T cell function or decreased numbers of CD4 cytotoxic T cells specific for EBV can trigger certain B cell malignancies or autoimmune diseases.
The references are extensive, however, I gave this section a low rating (only 3 stars) because the references are completely out of order. Something must have gone wrong with the reference manager program because the references cited in the manuscript are numbered incorrectly.
The figures are very busy and a bit confusing to follow. It may be helpful to number the events in the figure in the order that they are discussed in the figure legend.
Minor points:
p. 4, line 141: change, "individuals that is" to "individuals that are"
p. 8, line 350: change "germ center" to "germinal center"
p. 9, line 373: change, "with a cross-reaction to self-antigens" to "with cross-reactivity to self antigens"
p. 17, line 794; please rephrase this sentence, not sure what you are saying here because CD4 T cells cannot be cross-reacting with the TH1 response.
Author Response
All references have been reordered.
The events in each of the figures have been numbered in the order in which they are discussed, as suggested by the Reviewer. In addition, Figure 4 has been redesigned.
All minor points have been corrected:
p. 4, line 143: "individuals that are"
p. 8, line 353: "germinal center"
p. 9, line 376: "with cross-reactivity to self antigens"
p. 17, line 794: “By increasing the number of EBV-latent B cells due to this immunodeficiency by increasing the Th2 response, the presentation of viral antigens would increase, and consequently, there would be an increased risk of generating viral antigen-specific Th1 cells with cross-reactivity to self-antigens”.
Reviewer 2 Report
In this review, the author gives an overview of the immunological mechanisms associated with the development of EBV-related autoimmune diseases or cancer, proposes a model of EBV-associated autoimmunity or cancer development based on inflammation, formation of ectopic lymphoid structures, activation of CD4 T lymphocytes and self-antigen presentation through increasingly expressed MHC-II and, more importantly, provides a rationale for the use of novel agents to control EBV latency and development of EBV-related autoimmunity and cancer.
The review is very thorough and updated, and iconographic material helpful for the reader; however , the author should:
1. re-design Figure 4 and insert mechanisms for the other pharmacologic agents described in the discussion.
2. revise references, as the numbers in the text do not relate to the numbers of references in the list (as an example, ref. 283 mentioned in the text as a ref. for long COVID is a paper on gastric CA, while the ref. for long COVID is likely 288?; also 208 or 275, mentioned as ref. for chronic fatigue syndrome or myalgic encephalomyelitis are studies on T1D and T cell exhaustion, respectively)
Author Response
All references in the manuscript have been reordered.
The events in each of the figures have been numbered in the order in which they are discussed. In addition, Figure 4 has been redesigned.